# Coronal Knee Alignment and Tibial Rotation in Total Knee Arthroplasty: A Prospective Cohort Study of Patients with End-Stage Osteoarthritis

**DOI:** 10.3390/bioengineering11030296

**Published:** 2024-03-21

**Authors:** Andrej Strahovnik, Igor Strahovnik, Samo Karel Fokter

**Affiliations:** 1Valdoltra Orthopeadic Hospital, Jadranska Cesta 31, 6280 Ankaran, Slovenia; andrej_strahovnik@yahoo.com; 2Faculty of Medicine, University of Ljubljana, Kongresni trg 12, 1000 Ljubljana, Slovenia; istrah@gmail.com; 3Clinical Department of Orthopedic Surgery, University Medical Centre Maribor, Ljubljanska 5, 2000 Maribor, Slovenia; 4Faculty of Medicine, University of Maribor, Slomškov trg 15, 2000 Maribor, Slovenia

**Keywords:** knee osteoarthritis, total knee arthroplasty, femorotibial rotational mismatch, medial proximal tibial angle, external tibial rotation

## Abstract

Several studies have found a relationship between the rotational anatomy of the distal femur and the overall coronal lower limb alignment in knees with osteoarthritis (OA). Less is known about the rotation of the proximal tibia, especially in the context of total knee arthroplasty (TKA), where one of the goals of the surgery is to achieve the appropriate component-to-component rotation. The aim of this study was to investigate the relationship between the coronal alignment of the lower extremity and the relative proximal tibial rotation. A prospective cohort study of patients with an end-stage OA scheduled for TKA was conducted. All patients underwent a computed tomography (CT) scan and a standing X-ray of both lower limbs. A relative femorotibial rotation was measured separately for mechanical and kinematic alignment. A statistically significant correlation was found between the tibial varus and the external tibial rotation (*p* < 0.001). Out of 14 knees with high tibial varus (>5°), 13 (93%) and 7 (50%) knees had >10° of femorotibial rotation for the mechanical and kinematic alignment landmarks, respectively. In order to keep the component-to-component rotation within the 10° margin, more internal rotation of the tibial component is required in knees with higher tibial varus.

## 1. Introduction

Knee OA is a progressive joint disease and one of the leading causes of disability in elderly population [1]. It is characterized not only by cartilage degeneration and joint-space narrowing, but also by meniscal degeneration and tears, subchondral bone remodeling, infrapatellar fat pad inflammation with fibrosis, and synovitis, leading to the perception of knee OA as a whole-joint disease. Age, obesity, previous knee injury, joint malalignment, and instability are all strong risk factors for the development of knee OA. Many of these factors involve biomechanical and physiological influences which lead to abnormal joint loading and alteration in the structure, metabolism, and mechanical properties of joint tissues [2].

TKA is the standard treatment for the end-stage knee OA. It has become one of the most common surgical procedures, with an expected increased demand at a rate of 43% by the year 2050, reaching an incidence rate of 299 per 100.000 inhabitants [3]. At the same time, the incidence of revision procedures, which is a very technically demanding procedure, is expected to increase even more rapidly by almost 90% [3].

Internal malrotation of components has been one of the most common surgical errors in TKA [4]. Increased internal rotation of the tibial component has been shown to have a higher influence on the retropatellar pressure compared to femoral rotation [5]. In addition, the tension on the medial collateral ligament increases significantly with the internal rotation of the tibial component [6]. Several clinical studies have confirmed the tibial component rotation to be an important factor in the development of postoperative pain, knee stiffness, and patellar instability in patients after TKA [4,7,8,9,10].

The Insall line method uses the medial third of the tibial tuberosity and the insertion of the posterior cruciate ligament on the posterior border of the tibia as the reference points [11]. These reference points are accessible and widely used osseous markers for determining the intraoperative tibial component rotational alignment [12,13,14]. However, the position of the tibial tuberosity seems to vary, which might result in a significant malrotation between the femoral and tibial components in TKA [15,16,17,18,19]. Different methods for determining the intraoperative tibial rotational alignment have emerged (Akagi line, anterior tibial cortex, posterior tibial condylar axis, range of motion technique), but none has managed to prove the superiority [13,20,21,22,23,24,25]. The rotational position of the tibial component still relies on the surgeon’s preference and is, at best, a compromise between osseous anatomic and ligamentous soft-tissue input [26,27,28].

Compared to the proximal tibia, the rotational alignment of the distal femoral component in TKA is smaller and less variable [18]. Several studies have tried to explain the variability and have found a relationship between the rotation of the distal femur and coronal alignment of the lower extremity in patients with knee OA [29,30,31]. They have shown that, as the coronal alignment changes from varus to valgus the condylar twisting angle (CTA, the angle between femoral transepicondylar axis (TEA) and femoral posterior condylar line (PCLf)) increases. Much less is known about the relationship between the coronal alignment and the proximal tibial rotation.

While the TKA surgical technique emphasizes the component-to-bone rotation, the component-to-component rotation also needs to be aligned during the surgery. A rotational mismatch between femoral and tibial components (>10°) has been associated with poorer outcomes and pain after TKA [10,32,33]. Higher rotational mismatch can be expected in cases where preoperative rotational landmarks are already not aligned [34]. To complicate things even further, new alignment strategies have emerged in the last decades, with kinematic alignment aiming towards a more personalized component position [35,36,37]. While an argument for a constant, anatomically independent tibial component rotation can be made in mechanical alignment, a more personalized approach is warranted in kinematic alignment. The tibial component should therefore be placed not only in the anatomical coronal and sagittal positions, but also in the correct rotation. Nedopil et al. found an anterior-posterior axis of the oval boundary of the lateral tibial plateau to be a better marker for tibial component rotation, leading to a smaller femorotibial mismatch in a kinematically aligned TKA [38]. Most surgeons, however, are still more familiar with the tibial tuberosity and rely on this marker to determine their tibial rotation.

The objective of this research was to explore the correlation between the coronal alignment of the lower limb and the relative proximal tibial rotation in both mechanical and kinematic alignment strategy. The eventual existence of this relationship could help the surgeon to anticipate and more reliably position the tibial component into a more anatomically correct rotation, thus avoiding the rotational mismatch in TKA.

## 2. Materials and Methods 

### 2.1. Materials

After the appropriate approval from the national ethics commission (ref. no. 0120570/2020/4) was obtained and the study was pre-registered (ClinicalTrials.gov (accessed on 26 January 2024), ID NCT05295602), a prospective cohort study with patients scheduled for TKA was conducted. Patients with Kellgren–Lawrence stage III–IV OA were included and underwent TKA from March 2022 to December 2024. All patients signed an informed consent form informing them of the risk of radiographic exposure during a CT scan. The non-inclusion criteria were as follows: prior surgery or trauma that changed the hip and/or knee anatomy (hip or knee arthroplasty, femoral or tibial osteotomy, femoral or tibial fracture, tibial tuberosity transfer), fixed flexion contraction >15°, and immunological diseases of the knee (rheumatoid arthritis). In addition to OA knees scheduled for TKA, knees on the contralateral side (with or without OA) were also included if the non-inclusion criteria were fulfilled.

### 2.2. Methods

All participants underwent a CT scan of full-length lower limbs prior to a TKA. A CT scan was performed on IQon-Spectral CT (Philips Healthcare, Best, The Netherlands) from the upper L4 vertebra margin to the ankles using a tube voltage of 140 kVp and a slice thickness of 1.5 mm. All images were saved as Digital Imaging and Communications in Medicine (DICOM) files. Both knees were kept in neutral position during the scanning. The same protocol was followed in all cases. Three-dimensional reconstructions were performed from the CT data using the open source program 3D Slicer v5.3.0 [39]. In addition, a standard X-ray of the lower limb while standing was taken to measure the Kellgren–Lawrence grade and the hip–knee–ankle angle (HKA-X).

Two observers independently performed all measurements. In cases where the measured coronal angles differed by more than 1.5° (axial angles more than 3°), both observers made a second measurement together to reach an agreement.

The relevant points and axes were determined using the definitions previously described in the literature [40,41,42]. To minimize the human error, all important anatomical points were determined on the CT using a semi-automatic method (e.g., center of the hip joint was calculated using a best-fit sphere of manually selected points on the femoral head surface, the ankle joint center was calculated as a mid-point of the line connecting the manually selected tips of both malleoli). The coronal plane was defined as a plane through the hip center and PCLf. The following angles were measured in the coronal plane: mechanical medial distal femoral angle (MDFA), mechanical medial proximal tibial angle (MPTA), and angle between femoral mechanical and anatomical axes (FMA). The angles are shown in Figure 1. The arithmetical hip–knee–ankle angle (aHKA) was calculated by summing the MDFA and MPTA. To measure the rotational limb profile, the femoral torsion (tF, angle between femoral neck axis (FNA) and PCLf), CTA and tibial torsion (tT, angle between tibial posterior condylar line (PCLt), and ankle transmalleolar line (TMA)) were determined. When the direction of the distal line was in internal rotation, the angles were defined as positive.

The proximal tibial rotation was measured in relation to the distal femoral rotation, as this association is relevant for the purpose of TKA. A rotational femorotibial (rFT) angle was defined as an angle between distally projected TEA and the Insall line. In order to determine the Insall line, the medial third of the tibial tubercle was first projected proximally to the tibial plateau along the tibial mechanical axis. The Insall line was then defined as a line connecting the center of the posterior cruciate ligament and the projected medial third of the tibial tubercle. The TEA, connecting the sulcus of the medial femoral epicondyle to the lateral epicondyle, was projected onto the tibial plateau along the femoral mechanical axis. The angle between the projected TEA and the Insall line–rFT angle was then measured and used as an estimation for femorotibial rotational match (Figure 2). If the Insall line was externally rotated to the projected TEA, the angle was defined as negative. A rotational lateral plateau angle (rLAT), defined as an angle between the long axis of the lateral tibial plateau and the projected PCLf, was also determined, as this can be used as a measurement for components matching in kinematic alignment (Figure 3).

### 2.3. Statistical Analysis

All analyses were conducted using SPSS version 22 (IBM Corp., Armonk, NY, USA). Descriptive analyses were reported using means, standard deviations (SD), and ranges for continuous variables and frequencies with percentages for discrete variables. The relationship between the coronal and axial angles was analyzed with the Pearson correlation coefficient, and a significance of less than 0.05 was considered significant. The intraclass correlation coefficient was used as a measure of inter-rater reliability. A one-way ANOVA test was used to compare the differences among the Kellgren–Lawrence groups. Multiple linear regression was used to assess the impact of measured variables on rFT.

## 3. Results

One hundred ninety-nine knees (112 patients) were included in the study. A total of 25 patients had had a previous surgery on the other side (12 TKA, 11 total hip arthroplasties, 1 hip osteosynthesis after a fracture, and 1 patient with the transfer of tibial tuberosity); these previously operated knees were not included in the study. The demographic data are shown in Table 1.

A high degree of reliability was found between both observers. The intraclass correlation coefficients for relevant measurement are shown in Table 2.

A significant correlation between the overall limb alignment (aHKA) and the tibial rotation (rFT) was found (*p* < 0.001, Table 3).

An even stronger correlation (*p* < 0.001) was observed between the tibial varus (MPTA) and rFT, as shown in Figure 4. A collinearity existed between aHKA and MPTA, and the latter was found to be an independent parameter. Multiple linear regression (stepwise method) found MPTA, the slope of the medial tibial plateau, and internal femoral torsion (tF) to be meaningful predictors (rFT = −115.98 + (1.06 × MPTA) + (0.20 × medial tibial plateau) + (−0.10 × tF); adjusted R^2^ = 0.48, *F* _3, 195_ = 61.78, *p* < 0.001). 

In case of mechanical alignment strategy (using the Insall line), rotational femorotibial mismatch (rFT) of more than 10° was found in 41 knees (21%). In case of kinematic alignment, the long axis of the lateral tibial plateau was used (rLAT), and only 16 knees (8%) had rotational femorotibial mismatch outside the 10° margin. Thirteen out of fourteen (93%) patients with MPTA lower than 85° had the rFT > 10°. In patients with MPTA < 87°, the rFT > 10° was found in 27 out of 59 (46%) cases. Only 2 knees (both in one patient) out of 55 (4%) with MPTA > 90° had rFT higher than 10°. If the Insall line was measured for kinematic alignment (using PCLf instead of TEA), 117 knees (59%) had rotational femorotibial mismatch > 10°.

A one-way ANOVA was performed to compare the effect of Kellgren–Lawrence grade on rFT. No statistically significant difference in rFT was found among the groups (*F*
_2, 196_ = 1.07, *p* = 0.35). In addition, no correlation was observed between the joint line convergence angle on a standing long-leg X-ray and rFT (*p* = 0.09).

A rotational profile of a lower extremity with measured lines referenced to the FNA was analyzed. With increasing varus of the proximal tibia (lower MPTA), more internal rotation of TEA, PCLf, and PCLt was observed when referenced to FNA (*p* < 0.001, *p* < 0.001 and *p* = 0.003, respectively). No significant correlations of Insall’s line (*p* = 0.07, trend toward external rotation) and TMA (*p* = 0.69) were found as the MPTA changed.

Women had significantly higher internal femoral torsion and higher CTA (*p* = 0.002 and *p* = 0.029, respectively). No influence of gender on other relevant angles was found.

## 4. Discussion

The primary objective of this study was to find a relationship between the coronal lower limb alignment and the relative femorotibial rotation in patients scheduled for TKA. A significant correlation between the tibial varus (MPTA) and the relative femorotibial rotation (rFT) was found. Our statistical regression model showed that with more tibial varus, more external rotation of the proximal tibia could be expected (higher femorotibial rotational mismatch).

A similar relationship has already been described by Matsui et al. [43]. They demonstrated that relative external rotation of the tibia existed in knees with OA and was higher in knees with increased overall limb varus. Likewise, Khan et al. also found increased external tibial rotation in varus osteoarthritic knees [44]. Both studies used the tibial tuberosity as a marker for the proximal tibial rotation. If the tibial posterior condylar line was used instead, no increase in external rotation was found in either study. No statistically significant difference for PCLt was found in our study. The possible explanation for the discrepancy between the tibial tuberosity and PCLt is the formation of the osteophyte at the posterior rim of the medial tibial plateau, which typically forms in a varus knee. In our study, it was the tibial varus (MPTA) rather than the overall limb varus that was independently correlated to the proximal tibial external rotation.

In addition to the tibial varus, the internal femoral torsion (angle between the femoral neck and posterior condylar line of the femur) was correlated to the external tibial rotation. With lower MPTA, higher internal femoral torsion (higher anteversion) was observed. We believe that this relationship plays a role in the development of external tibial rotation. In a study by Nejima et al., a similar correlation was found between increased femoral torsion and lower MPTA [45]. They postulated that this relationship is a result of nature trying to restore the extension-to-flexion change in joint line obliquity, thus making the joint line parallel to the ground in a flexed knee. In order to maintain a normal foot progression angle with increased internal femoral torsion, a compensatory external rotation of the tibia may reduce the external knee’s adduction moment [46]. Increased femoral neck anteversion also shifts the mechanical axis towards valgus and decreases the stress across the medial compartment [47,48]. The relative external rotation of the tibia could therefore be a compensatory mechanism to decrease the growing compressive forces on the medial compartment as the OA progresses. Further confirmation of this theory was given in a cadaveric study by Yazdi et al. [49]. They found a decreased contact pressure of the medial compartment when the tibia was placed at 15° of external rotation (if the tibia was rotated interiorly, the pressure increased). If this theory is true, lower MPTA would, therefore, through increased femoral torsion, force the tibia into external rotation to alleviate the increasing medial compartment forces.

In addition to tibial varus and femoral torsion, the increased medial tibial slope was also found to independently correlate to the femorotibial rotation. The effect of increased posterior slope on anterior tibial translation has already been described by Nagamine et al. [50]. With an increased medial tibial slope, more anterior translation of the medial tibia can be expected, which would result in external rotation. To our knowledge, we are the first to report the connection between tibial rotation and medial tibial slope.

The knowledge of preoperative rotational femorotibial mismatch could have an important clinical implication for TKA. The conventional use of bony landmarks for determining the tibial component rotation might lead to rotational mismatch, especially in knees with increased tibial varus (MPTA < 85°). A method which includes a relative femorotibial rotational measure (e.g., range of motion technique, use of a connecting instrument) would probably be better suited to these situations [51].

No association between the Kellgren–Lawrence grade and external tibial rotation was found in our study. This suggests that rotation might develop before end-stage symptomatic knee OA. However, our study was not designed to follow patients through different stages of osteoarthritis, as we included only patients scheduled for TKA. On the other hand, there are studies which claim that the femorotibial rotation changes with the progression of OA [44,52]. In these studies, patients with a higher grade of OA (Kellgren–Lawrence) had higher external rotation of the tibia relative to the femur. Rotation could be a part of changes that affect the proximal tibia as the OA progresses. At the initial stage, the proximal tibial parameters, including the varus, remain more or less unchanged [53]. As the OA progresses, the “non-uniform settlement phenomenon” occurs, initially creating a medial shift of the femoral condyles (medial subluxation), followed by the bone loss of the medial tibial compartment (medial plateau settlement) and distinct morphological changes in the lateral compartment of the knee [54,55]. External tibial rotation could, therefore, be a part of this settlement phenomenon. The relative increase in femorotibial rotation would, in this case, be pathological, and following the preoperative femorotibial rotation during TKA, an attempt to recreate the pathological condition would be made.

The main question is whether the femorotibial mismatch corrects itself with TKA. If the external tibial rotation is a part of the end-stage osteoarthritic changes, the tibia might rotate back once those changes are corrected during TKA. Kawaguchi et al. looked at 79 patients with pre- and postoperative CT scans [34]. They found the existent preoperative tibiofemoral mismatch to be the most important risk factor for the remaining postoperative component rotational mismatch in TKA. This implies that, in knees with preoperative tibiofemoral mismatch (e.g., in knees with higher tibial varus), the rotation does not correct itself with TKA, and standard landmarks for component rotation might not be appropriate.

This is especially true in kinematic alignment, where placing the femoral component in a flexion–extension axis means placing it into more internal rotation. In order to keep the appropriate component rotation, the tibial component should also be rotated into a higher internal rotation, particularly in knees with more tibial varus, where more external rotation is expected. In kinematic alignment, the lateral plateau line has been suggested as a landmark for the tibial component rotation [38]. We have found this line to be an appropriate landmark in 92% of knees in kinematic alignment. That is slightly less than the findings of original authors (97%), but still much better if the Insall line is used for either kinematic or mechanical alignment. Although Insall’s line has been shown to be a reliable landmark in non-arthritic knees, that has not been proven in the population with end-stage OA [56]. We, therefore, suggest selecting an axis in the middle between the range-of-motion technique and Insall line as a reference for tibial component rotation in knees with a higher grade of tibial varus. This seems to be a sensible compromise for determining a better rotation of the tibial component in mechanical alignment. The alternative landmark, long anterior–posterior axis of the lateral tibial plateau, has been shown to be more reliable with less femorotibial rotational mismatch in kinematic alignment.

By observing the rotational profile of the whole limb with the femoral neck axis (FNA) as a reference line, an internal rotation of TEA and PCLf was found to be in strong correlation to higher tibial varus. This internal torsion of the distal femur seemed to be extra-articular at the level of the femoral diaphysis. A corresponding external rotation of the tibia was observed, and was intra-articular in nature (at the level between the distal femur and the proximal tibia). As a consequence, FNA–Insall line rotation and whole-limb rotation (FNA-TMA) did not correlate to the tibial varus and remained relatively constant to FNA. This further substantiates the theory that lower limb rotation attempts to maintain a normal foot progression angle regardless of aHKA and MPTA.

The study is certainly not without limitations. Firstly, it was conducted on patients who were in the pre-operative stage, scheduled for TKA, and the actual surgery had not yet been performed. Consequently, there was no opportunity to validate our findings in vivo, as there were no postoperative X-rays or CT scans available for analysis. Secondly, the assessment of OA was based solely on Kellgren–Lawrence staging, without considering factors such as patients’ pain levels, ranges of motion, and other relevant criteria. Thirdly, the CT scan was performed under the non-weight-bearing condition, and the results of rotational mismatch showed the non-weight-bearing status. The conclusions regarding our tibiofemoral rotational mismatch could, therefore, not be extended to weight-bearing conditions. Fourth, our proposal regarding the rotational position of the tibial component has not undergone clinical validation. Subsequent studies will be necessary in order to address these aspects comprehensively.

## 5. Conclusions

This study showed a correlation between tibial varus and external rotation of the tibia. Knees with more than 5° of tibial varus (MPTA < 85°) are highly likely to have an increased external tibial rotation and a pre-operative rotational femorotibial mismatch. The use of standard tibial rotational landmarks in these cases requires more internal rotation of the tibial component to avoid an excessive post-operative rotational component mismatch.

## Figures and Tables

**Figure 1 bioengineering-11-00296-f001:**
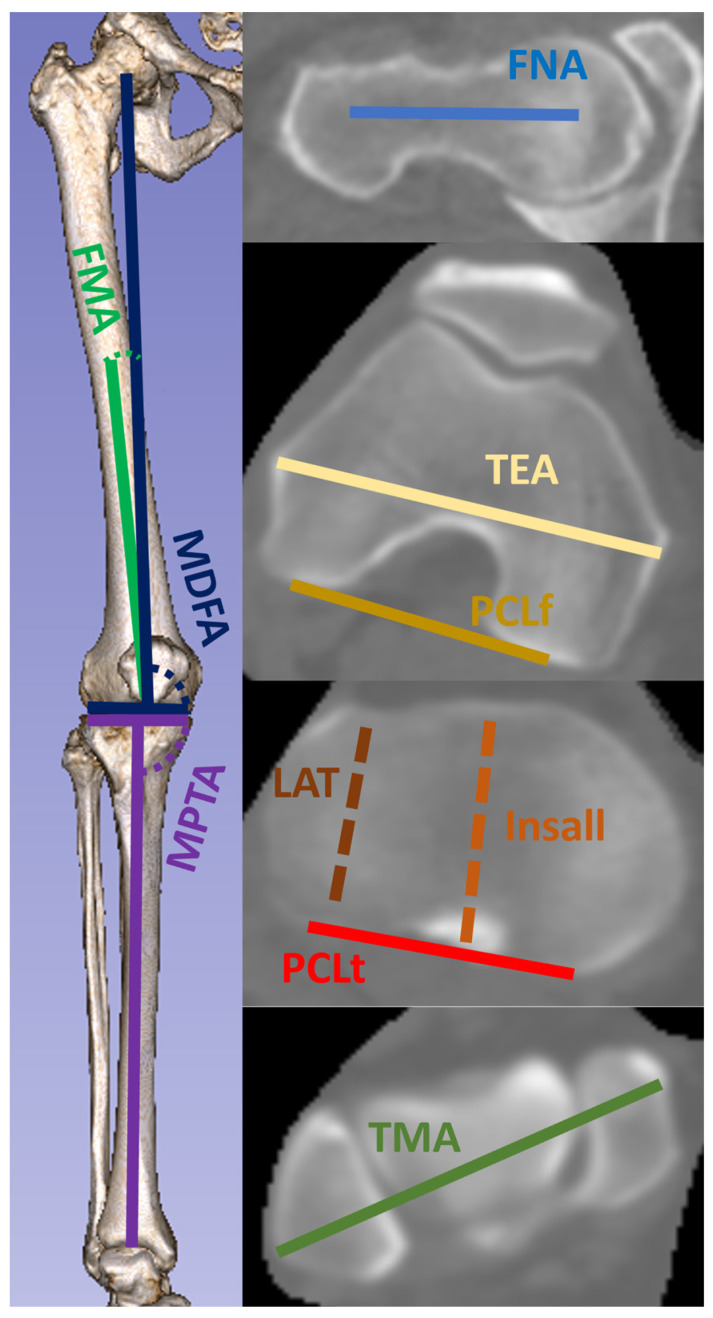
Schematics of main lines and measured angles. Correlations of angles between FNA and relevant lines to increasing varus of aHKA and MPTA are signified with *p* values. The direction (internal rotation) of significant correlations is marked with curved arrow. MDFA—mechanical medial distal femoral angle. MPTA—mechanical proximal tibial angle. FMA—femoral mechanical anatomical angle. FNA—femoral neck axis. TEA—transepicondylar axis. PCLf—femoral posterior condylar line. LAT—projection of anterior–posterior axis of lateral tibial plateau. Insall—projection of Insall line. PCLt—tibial posterior condylar line. TMA—transmalleolar axis. aHKA—arithmetic hip–knee–ankle angle.

**Figure 2 bioengineering-11-00296-f002:**
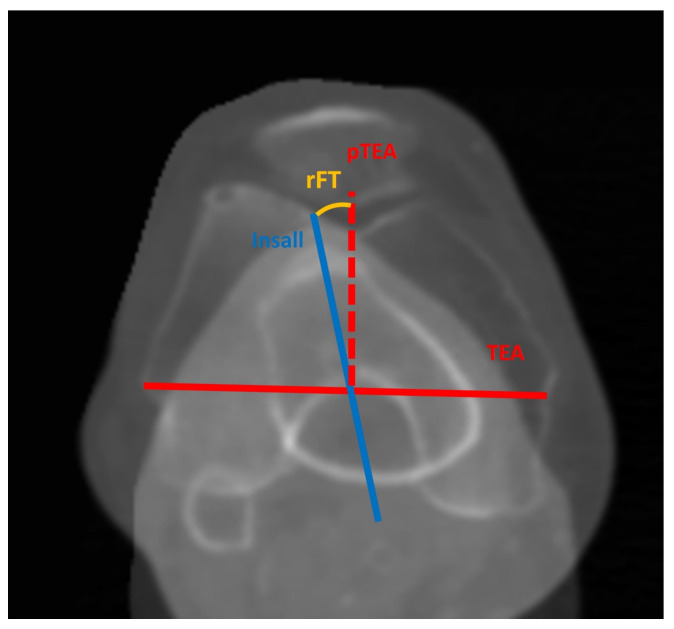
Measurement of rotational femorotibial angle (rFT) on superposed images of distal femur and proximal tibia. TEA—projection of transepicondylar line; Insall—projection of Insall’s line; pTEA—perpendicular line on TEA. If pTEA is in internal rotation to Insall line, values are positive.

**Figure 3 bioengineering-11-00296-f003:**
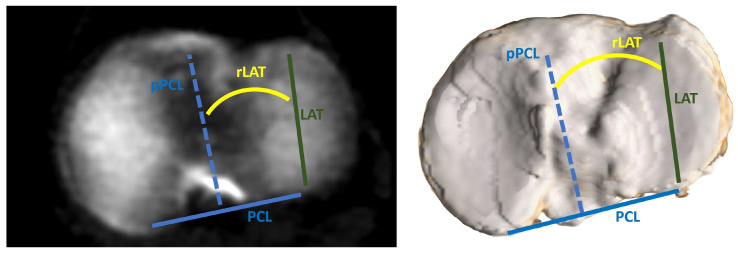
Measurement of rotational lateral plateau angle (rLAT). **Left**: transversal CT slice at the level of tibial plateau. **Right**: 3D volume rendering for better visualization of lateral tibial plateau long axis. PCL—posterior condylar line of the femur; LAT—long axis of lateral tibial plateau surface; pPCL—perpendicular line to PCL.

**Figure 4 bioengineering-11-00296-f004:**
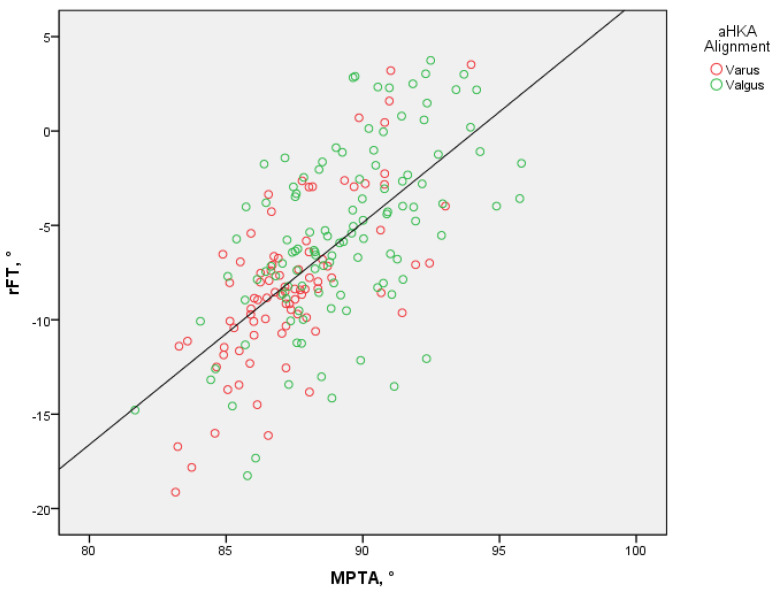
Relationship between rotational femorotibial angle (rFT) and mechanical medial proximal tibial angle (MPTA). aHKA—arithmetic hip–knee–ankle angle.

**Table 1 bioengineering-11-00296-t001:** Patient demographics and radiographic/CT data. HKA-X—hip–knee–ankle angle on a standing lower limb X-ray. aHKA—anatomical hip–knee–ankle angle on a CT scan. MDFA—mechanical medial distal femoral angle. MPTA—mechanical medial proximal tibial angle. FMA—femoral anatomical mechanical angle. CTA—condylar twisting angle. tF—femoral torsion. tT—tibial torsion. rFT—rotational femorotibial angle (mechanical alignment, based on the tibial tuberosity). rLAT—rotational lateral plateau angle (kinematic alignment, based on the lateral tibial plateau).

Patients, *n*	112
Knees, *n*	199
Age, years	69.7 ± 7.9 (49–84)
Gender, female/male	111/88
Side, right/left	100/99
Kellgren–Lawrence, n-grade 2/3/4	20/67/112
HKA-X, °	176.2 ± 7.0 (160.4–197.6)
aHKA, °	181.2 ± 4.1 (171.0–191.9)
MDFA, °	92.8 ± 2.5 (86.6–100.7)
MPTA, °	88.5 ± 2.6 (81.7–95.8)
FMA, °	6.1 ± 1.5 (2.6–11.4)
CTA, °	4.2 ± 1.7 (0.6–8.2)
tF, °	12.3 ± 7.7 (−13.2–34.2)
tT, °	−25.9 ± 8.4 (−57.5–-4.4)
Medial tibial slope, °	82.8 ± 3.8 (70.7–92.7)
Lateral tibial slope, °	84.1 ± 3.7 (72.3–95.2)
rFT, °	−6.7 ± 4.7 (−19.1–3.7)
rLAT, °	−6.5 ± 3.3 (−13.0–3.8)

Data are presented as means ± standard deviations with ranges in parentheses.

**Table 2 bioengineering-11-00296-t002:** Intraclass correlation coefficients (ICC) for relevant variables. HKA-X—hip–knee–ankle angle on a standing lower limb X-ray. aHKA—anatomical hip–knee–ankle angle on a CT scan. MDFA—mechanical medial distal femoral angle. MPTA—mechanical medial proximal tibial angle. FMA—femoral anatomical mechanical angle. CTA—condylar twisting angle. tF—femoral torsion. tT—tibial torsion. rFT—rotational femorotibial angle (mechanical alignment, based on the tibial tuberosity). rLAT—rotational lateral plateau angle (kinematic alignment, based on the lateral tibial plateau).

	ICC	95% Confidence Interval
HKA-X	0.99	0.98–1.0
aHKA	0.99	0.99–0.99
MDFA	0.99	0.99–1.0
MPTA	0.99	0.98–1.0
FMA	0.99	0.99–0.99
CTA	0.87	0.83–0.90
tF	0.97	0.97–0.98
tT	0.98	0.97–0.98
rFT	0.93	0.91–0.95
LAT	0.91	0.88–0.93

**Table 3 bioengineering-11-00296-t003:** Correlation between coronal and axial parameters. HKA-X—hip–knee–ankle angle on a standing lower limb X-ray. aHKA—anatomical hip–knee–ankle angle on a CT scan. MDFA—mechanical medial distal femoral angle. MPTA—mechanical medial proximal tibial angle. FMA—femoral mechanical anatomical angle. rFT—rotational femorotibial angle (mechanical alignment, based on tibial tuberosity). rLAT—rotational femorotibial angle (kinematic alignment, based on lateral tibial plateau). tF—femoral torsion. tT—tibial torsion. CTA—condylar twisting angle.

	rFT	rLAT	tF	tT	CTA
HKA-X	**0.21, 0.005**	0.14, 0.05	0.04, 0.56	−0.05, 0.48	0.04, 0.55
aHKA	**0.30, <0.001**	**0.23, 0.001**	−0.07, 0.35	−0.01, 0.84	−0.01, 0.99
MDFA	0.09, 0.21	0.04, 0.62	0.02, 0.74	**−0.22, 0.002**	**0.32, <0.001**
MPTA	**0.66, <0.001**	**0.44, <0.001**	**−0.29, <0.001**	**0.21, 0.003**	−0.07, 0.35
FMA	**−0.18, 0.01**	**−0.15, 0.04**	0.06, 0.39	0.04, 54	−0.10, 0.16

Data are presented as Pearson correlation coefficients (ρ) and statistical significance values (*p*, <0.05 values in bold).

## Data Availability

The anonymized Excel table can be obtained upon request from the corresponding author (privacy reason).

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
