# Peer review of "Coronal Knee Alignment and Tibial Rotation in Total Knee Arthroplasty: A Prospective Cohort Study of Patients with End-Stage Osteoarthritis"

_bioengineering, 2024, doi:10.3390/bioengineering11030296_

Round 1

Reviewer 1 Report

Comments and Suggestions for Authors

There is no introduction about knee OA and biomechanics of knee. It should be added that OA is a whole joint disease involving all joint tissues and not only cartilage degeneration as reported by the authors. OA is characterized by subchondral bone remodelling, meniscal degeneration, inflammation and fibrosis of both infrapatellar fat pad and synovial membrane. Risk factors should also be described.

Lines 31-33: references should be provided.

Lines 38-40: in OA patients?

Lines 45-46: in OA patients? in healthy people?

Lines 52-53: in OA or healthy people?

In general, the authors should specify when they are referring to data obtained from OA patients or healthy subjects.

Section 2: it should be reported the type of study.

Section 2 should be divided in subsections.

Line 85: number of patients enrolled should be moved to the results.

Lines 86: dates should be checked as it would be difficult to have included patients between March 2022 and December 2024.

Line 101: inter and intra-reliability should be added.

Line 102: 1.5Ëš° unit of measurement should be checked as there are two “Ëš°”

Figure 1 should be modified. P-values should be deleted as this section is related to methods and not results.  

Did the authors check distribution of the data before performing statistical analysis?

Section 3: flow-chart of patient’s selection is lacking.

Table 1: it is reported that 20 patients had Kellgren-Lawrence grade II. First, it is unclear why these patients were enrolled considering that the inclusion criteria (line 85) are that patients had to have Kellgren-Lawrence (KL) stage III-IV OA. Second, normally patients with KL 2 are not subjected to TKA.

Lines 185-186: the authors refer to Multiple linear regression (stepwise method). However, this test was not mentioned in the statistical analysis section. The same for one-way ANOVA at line 201.

Lines 201-204: this analysis should be checked as it is unclear how many patients were enrolled and KL grade.

Lines 205-207: here, the authors should report figure 1 with p-values.

It should be checked if gender has an influence on the correlations etc.

Lines 209-210: it should be specified “in knee OA patients”.

References should be placed between square brackets and not round brackets.

Reviewer 2 Report

Comments and Suggestions for Authors

Review

Journal

Bioengineering (ISSN 2306-5354)

Title:”Coronal Knee Alignment and Tibial Rotation in Total Knee Arthroplasty: A Prospective Cohort study of Patients with End-Stage Osteoarthritis” 

The objective of this research was to explore the correlation between the coronal 76 alignment of the lower limb and the relative proximal tibial rotation in both mechanical 77 and kinematic alignment strategy. The eventual existence of this relationship could help 78 the surgeon to anticipate and more reliably position the tibial component into a more 79 anatomically correct rotation, thus avoiding the rotational mismatch in TKA.

 It is well described article with proper method and conclusion. It is acceptable 

Title

Good

Abstract

Good

Methods

Good.

Results

Good.

Discussion

Good.

References

Good.

Round 2

Reviewer 1 Report

Comments and Suggestions for Authors

No additional comments